# Use of surveys to evaluate an integrated oral cholera vaccine campaign in response to a cholera outbreak in Hoima district, Uganda

Godfrey Bwire,[1] Mellisa Roskosky,[2] Anne Ballard,[2] W Abdullah Brooks,[2] Alfred Okello,[3] Florentina Rafael,[4] Immaculate Ampeire,[5] Christopher Garimoi Orach,[3] David A Sack  [2]

[1]Department of Community Health, Ministry of Health, Kampala, Uganda
[2]Department of International Health, Johns Hopkins Bloomberg School of Public Health, Baltimore, Maryland, USA
[3]Makerere University, College of Health Sciences, Kampala, Uganda
[4]Department of Infectious Hazard Management, World Health Organization, Geneva, Switzerland
[5]Ministry of Health, Uganda National Expanded Program on Immunization, Kampala, Uganda

**Correspondence to**
Dr David A Sack;
dsack1@jhu.edu

## ABSTRACT

**Objectives** To evaluate the quality and coverage of the campaign to distribute oral cholera vaccine (OCV) during a cholera outbreak in Hoima, Uganda to guide future campaigns of cholera vaccine.

**Design** Survey of communities targeted for vaccination to determine vaccine coverage rates and perceptions of the vaccination campaign, and a separate survey of vaccine staff who carried out the campaign.

**Setting** Hoima district, Uganda.

**Participants** Representative clusters of households residing in the communities targeted for vaccination and staff members who conducted the vaccine campaign.

**Results** Among 209 households (1274 individuals) included in the coverage survey, 1193 (94%; 95% CI 92% to 95%) reported receiving at least one OCV dose and 998 (78%; 95% CI 76% to 81%) reported receiving two doses. Among vaccinated individuals, minor complaints were reported by 71 persons (5.6%). Individuals with 'some' education (primary school or above) were more knowledgeable regarding the required OCV doses compared with non-educated (p=0.03). Factors negatively associated with campaign implementation included community sensitisation time, staff payment and problems with field transport. Although the campaign was carried out quickly, the outbreak was over before the campaign started. Most staff involved in the campaign (93%) were knowledgeable about cholera control; however, 29% did not clearly understand how to detect and manage adverse events following immunisation.

**Conclusion** The campaign achieved high OCV coverage, but the surveys provided insights for improvement. To achieve high vaccine coverage, more effort is needed for community sensitisation, and additional resources for staff transportation and timely payment for campaign staff is required. Pretest and post-test assessment of staff training can identify and address knowledge and skill gaps.

## INTRODUCTION

Cholera, a preventable and treatable disease, is characterised by profuse watery diarrhoea caused by infection of the intestine with the bacterium *Vibrio cholerae*.[1] Cholera is a major cause of morbidity and mortality

### Strengths and limitations of this study

► The cluster survey of households in communities targeted for vaccination efficiently documented actual vaccine coverage in the target population.
► The cluster surveys of households identified mild adverse events not identified during the campaign and identified the need to emphasise the second dose, especially among less educated groups.
► Surveys of the vaccination staff immediately following each round identified certain weaknesses in staff orientation as well as constraints to their job performance in the field.
► The household surveys obtained data from a single spokesperson for the household rather than from each individual which might have introduced some uncertainty in the household data.
► Evaluation of the vaccination staff was carried through surveys and would have benefited by direct observation of the training and the field performance.

in several countries in sub-Saharan Africa where cholera outbreaks also negatively affect development due to associated high economic burden.[2 3] Between 2010 and 2016, an average 141 918 incident cases annually were reported from sub-Saharan African countries, including Uganda.[4] In Uganda, cholera outbreaks occurred as both endemic and epidemic disease. Epidemic disease occurred in northern and eastern Uganda districts[5] and are thought to be worsened by contamination of water due to poor sanitation.[6] Cholera outbreaks especially occur in districts along the international borders with the Democratic Republic of the Congo (DRC), South Sudan and Kenya and along the Great Lakes.[5 7] These districts include Hoima, where cholera is endemic.[5 8–10]

There has been debate in the public health community on best practices for endemic and epidemic cholera disease control, with

some preferring to focus on water, sanitation and hygiene (WASH) interventions, and others advocating for oral cholera vaccine (OCV) for both endemic and epidemic disease control.[11] In part, this has been facilitated by a relative lack of experience with OCV and concern that excess reliance on vaccine might negatively affect essential infrastructural development and hygienic practices. WHO recommends an integrated approach to cholera prevention where WaSH interventions are complemented by vaccine campaigns which provide OCV to persons living in areas considered high risk.[2 12] These vaccine campaigns may be either preventive, in which the vaccine is targeted to cholera hotspots, or reactive in which the campaign is implemented in response to an outbreak or a humanitarian emergency.[13]

Two WHO-prequalified currently OCVs are available from the global stockpile: Shanchol (Shantha Biotechnics, India) and Euvichol (Eubiologics, Korea).[2] The standard immunisation schedule consists of two doses given at an interval of at least 2 weeks to all persons in the target area above 1 year of age. While there is increasing use of OCV to control outbreaks, preventive use is constrained due to inadequate vaccine supply.[14] Since creation of a global OCV stockpile in July 2013, several OCV campaigns had been successfully implemented[13 14] but it is still important to document national campaign experiences as well as monitoring and evaluation activities, to continually improve the effectiveness and efficiency of vaccine campaigns.

The Ugandan Ministry of Health (MoH) had prepared plans for OCV campaigns in the areas identified as cholera hotspots starting in the western districts of Uganda (including Hoima), near the border with DRC and close to, or adjoining Lake Albert. These hotspot districts and their specified subcounties were confirmed during a national cholera workshop in Kampala on 29 January 2018–31 January 2018. This workshop led to the development of an application for OCV to the Global Taskforce for Cholera Control which was submitted on 14 February 2018. The application proposed providing OCV in these identified hotspots as a preventive strategy. However, while preparations for these campaigns were underway, an outbreak was declared in Hoima district on 23 February 2018. The earliest cases were identified among DRC refugees, but then other cases were seen among the non-refugee Ugandan population. The MoH responded to the outbreak with multisectoral interventions, including proper case management, promotion of access to safe water and improved sanitation (WaSH), enhanced cholera surveillance, as well as infection control and health education. These measures were then supplemented with plans for an emergency OCV campaign. Thus, the original plans for a preventive OCV campaign were shifted to an emergency response to control the outbreak. The first doses of vaccine arrived on 28 March and the first round of vaccinations started on 2 May. The doses for the second round arrived on 29 May and the second round started on 26 June. A door-to-door strategy

was used to deliver two doses of vaccine to an estimated 360 000 people, including pregnant women, over the age of 1 year residing in the four targeted subcounties.

To carry out the campaign, the MoH organised all activities including logistics, community mobilisation and implementation, coordinating ground activities through an assigned point person. Many stakeholders contributed to the campaign including the Hoima district local government, WHO, UNICEF, UNHCR and Médecins sans Frontiers. Prior to the campaign, the stakeholders met to define and coordinate their complementary tasks.

The epidemic curve based on a line list of cases and deaths by date and stated nationality is shown on figure 1. Over the course of the outbreak, 2122 cases with 44 deaths (case fatality rate (CFR), 2.1%) were reported. Sixty-six per cent (1410) of the cases and 64% (28) of the deaths occurred during the first 2 weeks of the outbreak. Many of the cases and deaths (1276 and 32, or 60% and 73%, respectively) occurred among persons who were from DRC, and the refugees developed cholera symptoms soon after arrival in Uganda. Among the 44 deaths reported, 25 (57%) occurred in the community, not in the health facility. Nineteen of the fatal cases were treated at the health facility; the CFR for facility-treated patients was 0.9%. Although the emergency vaccination campaign intended to control the outbreak, because the outbreak was so sudden and so short lived, the campaign could only be initiated after the outbreak had already declined.

## Rationale

While vaccines are commonly used in Uganda, especially through the longstanding programme (Expanded Programme on Immunisation), this was the first OCV use in Uganda, and there was no prior experience to guide responders and implementers. Thus, this study was carried out with the aim to document campaign activities and to monitor and evaluate its procedures and outcomes that could guide future OCV campaigns. The issues addressed during this study included the knowledge and practices (KP) of the campaign staff, vaccine coverage in the targeted areas, and the KP of the community. After this initial campaign, the MoH continued its plans for preventive campaigns in the remaining cholera hotspot districts, informed by the lessons learnt from this initial emergency use campaign. In an effort to document the impact of OCV on an outbreak in a setting with endemic disease, we undertook this monitoring and evaluation exercise, as described below.

## MATERIALS AND METHODS
### Study setting
Hoima district is located in western Uganda, across Lake Albert from DRC. It has a total area of 5735.5 km$^2$ and a projected population of 630 000 persons (2018). The district consists of 13 administrative units as follows: 10 subcounties (Kyabigambire, Buhimba, Kyangwali, Kabwoya, Bugambe, Kiziranfumbi, Kitoba, Kigorobya,

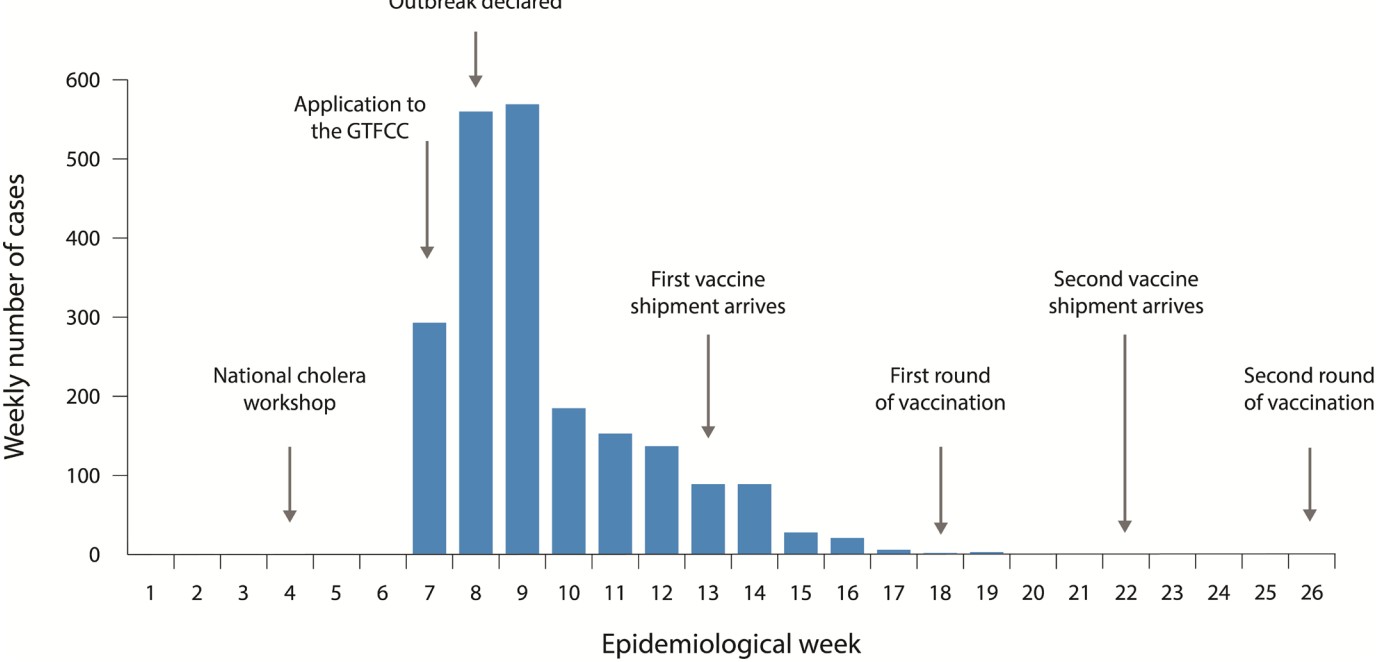

**Figure 1** Epicurve of the Hoima outbreak, 2018 with events identified in response to the outbreak. GTFCC, global taskforce for cholera Control.

Buseruka and Buhanika), a municipality (Hoima municipality) and two town councils (Kigorobya and Buhima town councils). The major economic activities of the population in Hoima are subsistence agriculture and fishing. Cholera is endemic in the district but the endemicity is localised in some specific subcounties particularly those with fishing communities.[15] The subcounties targeted for OCV included Buseruka, Kabwoya, Kangwali and Kigorobya which together constitute the six administrative units of Kyangwali, Kigorobya, Kabwoya Buseruka, Kigorobya town council and Kyangwali refugee settlement (Old and New) as shown in figure 2.

## Population and design for the monitoring and evaluation of the OCV campaign

Two substudies were conducted to assess different aspects of the campaign. In substudy 1, a representative sample of the population that was targeted for vaccination was questioned to determine vaccine coverage rates, detect adverse events following immunisation (AEFI) and collect additional information from the communities about the vaccine campaign. Substudy 2 consisted of a survey among the campaign staff who participated in the OCV campaign after each round to assess their KP.

## Substudy 1: community assessment

Substudy 1 was a two-stage, cluster survey conducted in the vaccine target area, consisting of 31 clusters, each cluster consisting of 4–7 households per cluster. The study population included each person >1 year of age who was living in the OCV targeted area at the time of the vaccination campaign. We assumed a household size of five persons based on estimates from a Demographic Health Survey conducted in 2016.[16] The sample was increased in order to raise the analytical power and precision of the surveys and to allow for separate analysis by gender. The formula used for determining sample size was, $n=(z^2pq)/d^2$,[17] where 'n' is the number of people desired for the survey, 'd' is the precision of the result, 'z' is the confidence limit, and 'p' and 'q' correspond to the proportion of persons in the population who are immunised and not immunised, respectively. We chose to use a low coverage of 50%.

To identify the clusters, a list of villages was obtained from each of the four sub-counties targeted for vaccination. From these lists, the Excel random number generator (=RANDBETWEENBOTTOM, TOP) was used to select the 31 villages from which households were selected. The number of villages per subcounty was proportionate to the population of the subcounty. The subcounty populations were obtained from the district planning unit.

From each selected village, a list of households was obtained from the village administrative leader (local council (LC)-1. This is the smallest recognised administrative unit in Uganda. It is headed an elected leader called LC-1) who provided a list of households from which we randomly selected households to interview.

For the household interviews, data were collected through standardised questionnaires during face-to-face interviews conducted by trained research assistants using the local language. Within a selected household the questionnaires were administered to the key respondents (head of the head of the households (HHs)), who represented the entire household and provided information about each member of the household. If a suitable key respondent was absent, additional visits were scheduled. In two households, a person could not be located, and

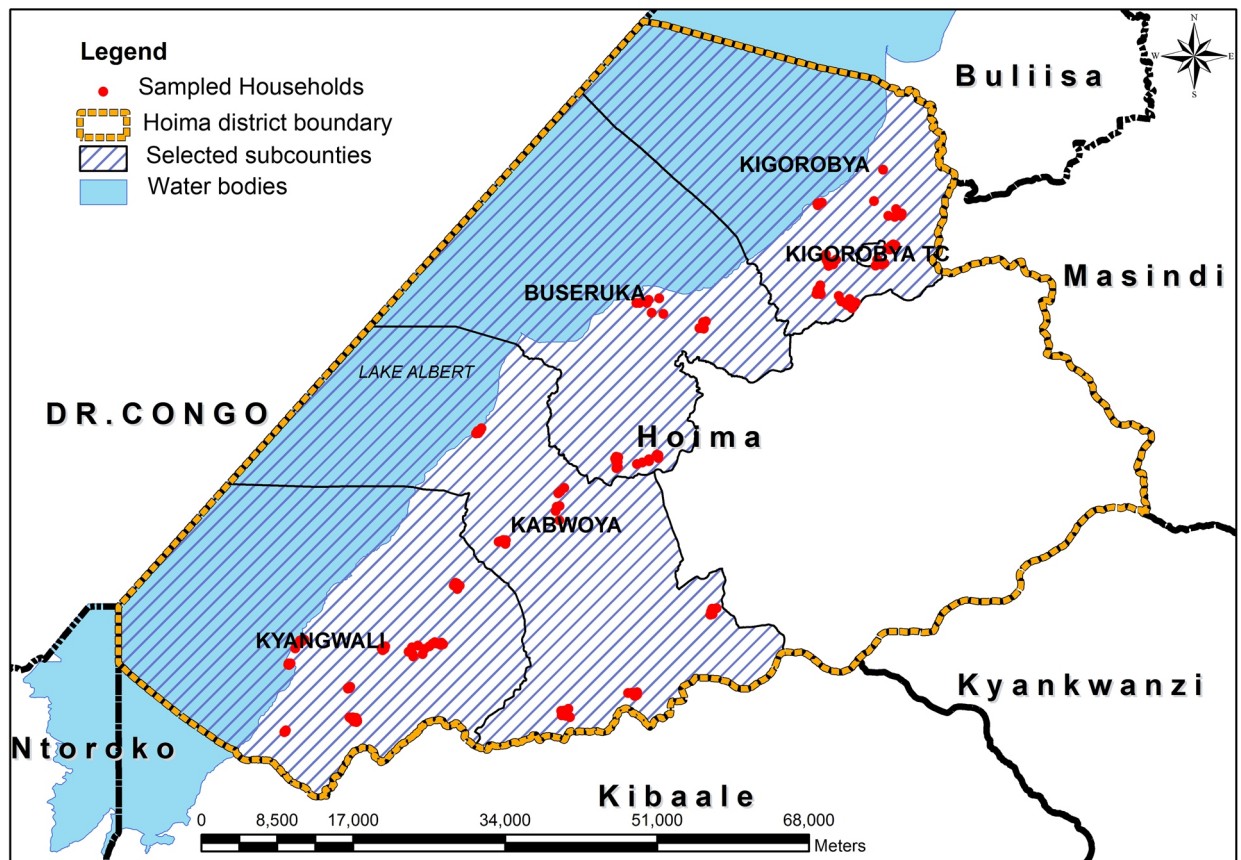

**Figure 2** Map of Hoima district, showing subcounties that received OCV and households where interviews were conducted (red dots). OCV, oral cholera vaccine.

the household was dropped. For each vaccinated person, the research assistants assessed cholera immunisation status. Vaccination status was ascertained in two ways: either the informant verbally indicated that the individual had received the dose of vaccine or the vaccination was recorded on the vaccine card. If the informant did not have a vaccine card, the reliability of the vaccination information was validated by asking about the details of the procedures of the vaccination (eg, being given recently by mouth to all persons >1 year of age). None of the residents who were approached refused to answer the survey. The age and sex of the participants in the survey is shown in table 1.

### Adverse events following immunisation

The occurrences of AEFIs were assessed by asking for symptoms among the vaccine recipients following vaccination.

**Table 1** Age and sex breakdown of the participants in the cluster survey

| Age group (years) | Male | Female | Total |
|---|---|---|---|
| 1–4 | 98 | 93 | 191 |
| 5–14 | 236 | 207 | 443 |
| 15–44 | 210 | 285 | 495 |
| 45+ | 80 | 65 | 145 |
| Total | 625 | 650 | 1274 |

As part of the campaign itself, a routine system for AEFI detection was established in which the vaccine team members advised vaccinees to report to a health worker or to seek care at a health facility if they experienced symptoms following immunisation. By contrast, the AEFI surveillance in this substudy asked the participants who participated in the cluster survey about symptoms they may have experienced. This AEFI substudy was, thus, designed to enhance our understanding of potential AEFIs which may not have been reported through the routine AEFI surveillance.

AEFIs were categorised for each individual member of the household who received a dose of OCV as follows. They were considered mild if the symptoms did not interfere with normal activities; moderate if they interfered somewhat; and severe if the symptoms prevented the individual from continuing normal activities.[18] Persons who reported to be having ongoing symptoms >72 hours were advised to visit the nearest health facility for more care. Among those reporting symptoms, information was recorded as to whether the person took any medicine or received any treatment to lessen the symptoms.

### Data collection and analysis

Data from the community surveys were collected by tablet computers using Kobo Collect (https://www.kobotoolbox.org/) installed to record the responses in the field. Data were cleaned, coded, and stored in Stata V.14. Data

were analysed to generate frequencies, percentages or proportions and means. Comparisons between groups was done via logistic regression for the calculation of ORs and 95% CIs. The results of analysis were presented in the form of graphs, tables, charts and means and were included in interim and end of campaign reports.

## Quality assurance

Research assistants were trained on data collection methods and were able to consult field supervisors and the principal investigator on any issue that was not clear to them. For quality assurance, the survey supervisors revisited about 10% of the households, not to collect the data again, but to ensure that they were not skipped by the interviewers for eligible respondents. The surveys were conducted about 2 weeks after completion of the second round of the vaccination campaign to minimise recall bias.

## Substudy 2: staff assessment

Staff were assessed on their level of knowledge on the cause of cholera, the importance of safe water in cholera prevention, the target age group for cholera vaccination, and knowledge about AEFI and the procedures for care should subjects experience an AEFI. Staff were surveyed twice with each survey taking place within 2 weeks after administration of the first and second OCV rounds, respectively. All staff who participated directly by administering the vaccines or indirectly through supervisory roles and who were present at the workstation during the study period were enrolled in the survey. For the staff survey, structured questions were administered on paper questionnaires that allowed for adding text to explain the answers (open-ended questions). Most of the vaccine staff had taken part in other public health campaigns, but none had participated earlier with a campaign to distribute OCV.

Written informed consent was obtained from all participants in both substudies. Participation in the study was voluntary and respondents were free to opt out at any stage of the interviews.

Confidentiality was observed at all stages of the study. No names or personal identifiers were included on the questionnaires. The research assistants underwent training on interview techniques, neutrality and research ethics. The benefits of the study to the staff included the ability to express themselves, provide feedback and observations

that in turn might lead to improvements in supportive services for their training and work.

## Patient and public involvement

This research was done without research subject involvement. The time was inadequate to involve the subjects prior to the vaccine campaign. They were also not invited to contribute to the writing or editing of this document for readability or accuracy. However, the findings of the study were disseminated to the Hoima district administration, MoH and other policymakers to use them to strengthen health service interventions and future OCV campaigns.

## RESULTS

### Substudy 1: community survey results

The community surveys were carried out in the four subcounties in Hoima districts of Buseruka, Kabwoya, Kyangwali and Kigorobya as shown in figure 2 and table 1. A total of 209 households, including 1274 individuals, were surveyed. Most (96%) of the respondents were household heads or their spouses. All respondents confirmed that they were living in the targeted OCV area at the time of the campaign. Fifty-one per cent of the respondents had primary education, 17% had secondary education, 1% had tertiary education and the remaining 31% had no education. The respondents were aged 18–89 years with a mean age of 40 years. Both sexes were present, with no statistically significant difference.

By verbal reports, 94% (95% CI 92% to 95%) of the residents received at least one dose and 78% (95% CI 76% to 81%) received two doses of OCV. From verbal reporting, 91% (95% CI 90% to 93%) of residents received vaccine during the first round and 81% (95% CI 78% to 83%) received vaccine during the second round. For many of the households, a vaccine card was available, and the vaccination card was used to confirm vaccination status. Using information from the card only, coverage was 84% (95% CI 82% to 86%) and 65% (95% CI 62% to 67%) for round 1 and round 2, respectively, and the two-dose coverage was 62% (95% CI 60% to 65%). Coverage rates are shown in table 2.

Among those who did not receive a dose of vaccine, over half of these missed doses (254 of the 357 missed doses during the two rounds) were because the person

**Table 2** Vaccination coverage post OCV campaign, Hoima district, Uganda, 2018

| Total surveyed=1274 | Round 1 | Round 2 | Received only one dose | Received two doses | Received at least one dose |
|---|---|---|---|---|---|
| Reported no (%) (95% CI) | 1164 (91.4) (89.7 to 92.8) | 1027 (80.6) (78.3 to 82.7) | 195 (15.3) (13.4 to 17.4) | 998 (78.3) (76.0 to 80.5) | 1193 (93.6) (92.2 to 94.9) |
| Confirmed by availability of the vaccination card (%) (95% CI) | 1065 (83.6) (81.5 to 85.5) | 823 (64.6) (61.9 to 67.2) | 142 (11.1) (9.5 to 13.0) | 792 (62.2) (59.5 to 64.8) | 934 (73.3) (70.8 to 75.7) |

OCV, oral cholera vaccine.

**Table 3** Treatment and resolution of adverse events following immunisation in Hoima district, Uganda, 2018

| Symptoms | Treatment | | Status | | |
|---|---|---|---|---|---|
| | No treatment (%) | Treated (%) | Recovered (%) | Ongoing (%) | Improved, not to baseline (%) |
| Mild | 24 (80.0) | 6 (20.0) | 29 (96.7) | 1 (3.3) | 0 (0.0) |
| Moderate | 11 (39.3) | 17 (60.7) | 23 (82.1) | 3 (10.7) | 2 (7.1) |
| Severe | 8 (61.5) | 5 (38.5) | 10 (76.9) | 0 (0.0) | 3 (23.1) |

was not at home at the time of vaccination or was out of town. In a few cases, the vaccine team missed the household, accounting for 53 missed doses. Refusing to take vaccine was not reported.

### Reported AEFIs

Overall, 71 individuals of 1274 respondents (5.6%) reported an AEFI (table 3). Determining a causal relation between the vaccination and the reported symptoms was not attempted.

Most AEFIs were considered mild or moderate, but 0.6%) persons reported an AEFI as severe. Most (60%) of the persons reporting an AEFI did not seek treatment including 60% of those reporting a severe AEFI. 29.6% of the reported adverse events occurred in the first round, 40.9% in the second round and 29.6% in both rounds. The most common symptoms were abdominal pain (15), diarrhoea, (9), fever, nausea and headache (each 6) reports. Table 4 provides additional information on the AEFIs. The reported AEFIs were infrequent relative to the number of doses distributed and there were no serious adverse events reported.

### Community knowledge of OCV

A majority (77%) of the respondents understood that vaccine was one of the ways to prevent cholera. There was a statistically significant association between education level and knowledge about OCV with those having at least a primary school education being almost twice as likely to know the number of required doses as compared with those with no education (OR 1.90, 95% CI 1.06 to 3.44) (p=0.03).

### Substudy 2: staff survey

A total of 242 and 125 staff responded to the first and second KP surveys (KP1 and KP2). Most respondents were vaccination team members (89% and 87% in vaccine rounds 1 and 2, respectively). Almost all the respondents were knowledgeable about the cause of cholera, the importance of safe water in cholera prevention and the vaccine target group, but were less knowledgeable regarding potential adverse events following administration (AEFI) or how to advise vaccinees, with 29% and 16% being less informed about AEFI during the first and second surveys.

**Table 4** Onset and frequency of symptoms reported as adverse events

| | <6 hours | 6–12 hours | 12–24 hours | 1–7 days | 8–14 days | Total |
|---|---|---|---|---|---|---|
| Diarrhoea | 3 | 2 | 2 | 2 | 0 | 9 |
| Vomiting | 3 | 0 | 0 | 2 | 0 | 5 |
| Nausea | 6 | 0 | 0 | 0 | 0 | 6 |
| Abdominal pain | 15* | 0 | 0 | 0 | 0 | 15 |
| Stomach gurgling | 0 | 3 | 0 | 1 | 0 | 4 |
| Mouth ulcers | 0 | 0 | 0 | 0 | 1 | 1 |
| Cough | 1 | 1 | 0 | 3 | 1 | 6 |
| Felt feverish | 1 | 1 | 0 | 6 | 1 | 9 |
| Poor appetite | 1 | 0 | 0 | 0 | 0 | 1 |
| Dizziness | 0 | 3 | 0 | 0 | 0 | 3 |
| Fainted | 0 | 1 | 0 | 0 | 0 | 1 |
| Itching | 0 | 0 | 0 | 0 | 1 | 1 |
| Weakness | 0 | 0 | 0 | 1 | 0 | 1 |
| Headache | 3 | 0 | 2 | 1 | 0 | 6 |
| Other | 1 | 0 | 0 | 2 | 0 | 3 |
| Total | 34 | 11 | 4 | 18 | 4 | 71 |

Three persons reported abdominal pain in one household.

When staff were asked to suggest areas that needed improvement in future OCV campaigns, more than 10% suggested more timely payment of allowances, more time to sensitise and inform the communities on the benefits of the vaccine, and better transportation and facilitation allowances (payments to health workers to cover the cost they incurred when administering the vaccines or conducting activities related to the OCV campaign). Other suggestions included use of both static and mobile vaccination points, provision of gumboots, umbrellas and more areas for vaccine storage in subcounties where vaccine would be more accessible, more workers for hard to reach areas, and an increase in the number of vaccine days to complete the vaccinations and increase coverage.

## DISCUSSION

The results of this monitoring and evaluation exercise documented important findings on the OCV campaign, the KP of both the community and the health staff involved in the campaign and implications for the conduct of future OCV campaigns as part of an integrated cholera control strategy. These findings suggest that the OCV campaign in Hoima successfully provided the vaccine to a very large proportion of the target population in Hoima district, western Uganda. Approximately 93.6% of respondents reported receiving at least one dose and 78.3% reported receiving two doses among residents. Given the mobile and transient nature of this population, this was noteworthy, and suggests that even better coverage may be possible for more settled populations in Uganda.

Since this was the first such campaign with OCV, there was concern that the population might be reluctant to accept it. This is a vaccine with which they were not familiar, it was given orally to all ages rather by injection to children, and two doses were required. Despite these potential constraints, we found that most people accepted taking the vaccine readily; however, some were not at home resulting in missed vaccinations.

High vaccination coverage is especially important when one is attempting to achieve herd protection. Since it is estimated that herd protection can be achieved with a coverage even lower than 90%[19]; the high coverage achieved in this campaign would be expected to induce significant indirect protection even among those who did not receive vaccine.[19 20]

It was noted in the administrative report from the MoH and during a stakeholder's meeting that one of the reasons for the reduction in the coverage during the second OCV dose was the unpredictable campaign dates for the second round. The vaccine for the second round had to be shipped and cleared through customs, and the timing for this was not certain. To avoid this problem in the future, a mechanism needs to be established to provide a better timeline for receipt of the vaccine shipments.

### Community reception to OCV

As with previous OCV campaigns outside Uganda, very few AEFI were reported.[21 22] Most of adverse events were considered mild or moderate and were self-limited. Despite the low prevalence of AEFI, the survey exposed the need to better inform the community about seeking treatment for more severe adverse events or for those that do not quickly resolve. This was especially true for families with little education who were less likely to seek medical attention for severe AEFI (data not shown). Notably, members in the community demonstrated good understanding of the rationale for the vaccine; however, a key takeaway from the survey was a need to better communicate the number of required doses, given that those with more education were twice as likely than those with no education to know the number of doses needed.

### Staff reception to OCV

Inclusion of staff KP survey contributed to the success of the project by identifying gaps among the staff knowledge and performance. Questioning the vaccine staff about their training and their experience in the field is not a common activity when conducting monitoring and evaluation activities during OCV campaigns. Many people had to be mobilised quickly and these were the key people who interacted with the communities. It was important that the staff accurately represent the campaign as an integrated cholera prevention programme, but this was the first time these people carried out this role. The MoH felt it important to monitor their knowledge and behaviours as well as any constraints they felt in carrying out their functions. While they were generally knowledgeable about the disease and about the vaccine, these staff needed additional training regarding recognising and managing AEFIs. They also faced challenges regarding logistical support. After the first staff KP survey, these gaps were communicated to the MoH so that appropriate actions could be taken to ensure that these gaps were addressed prior to the second round.

Most other OCV campaigns have also reported high coverage rates. These have included reports from Bangladesh,[23 24] Malawi,[25 26] Mozambique,[21] DRC,[27] Zambia,[28] South Sudan,[29 30] Iraq,[31] Haiti[32] and Guinea.[33] Clearly, OCV is well accepted among these very diverse population groups where the vaccine campaigns have been carried out.

Important limitations of this study need to be mentioned. Ideally, one would prefer to conduct community studies prior to a campaign to understand knowledge and attitudes about cholera to improve communications regarding the upcoming campaign as part of an integrated strategy to control cholera. However, since the campaign was carried out on an emergency basis during an outbreak, a study prior to the campaign was not possible. Second, a community survey immediately after the first round might have provided feedback to the teams that would have improved the coverage for the second round. It should also be noted that a single informant

provided information about receipt of the vaccine for all members of the household, so this informant might have incorrect information concerning one or more members of the household; however, since the vaccine was directly given to the household members together, it seems that inaccuracies would be minimal. It should be noted that we were not able to adjust for cluster sampling in the community surveys. If we have adjusted for cluster effect, it would have increased the variance slightly, but it would not affect the means.[34] The community KP survey did not include questions on attitudes regarding cholera. Since the survey had to be carried out very quickly following the campaign, and since the survey was targeted to identify issues that would be immediately relevant to campaign performance, it was felt that understanding attitudes regarding cholera, even though important, would have required other qualitative methods requiring more time than was available. Similarly, direct observation of the training and coordination meetings would have been useful to independently assess the efficiency and effectiveness of these training and coordination meetings. Furthermore, there was no list of all workers in the campaign and many of the workers who participated in the second round had left prior to administering the questionnaire; thus, there were fewer respondents in the second round and the proportion of all workers who participated could not be determined precisely. Finally, it was not possible, given the time constraints, to fully integrate WASH interventions together with the OCV campaign, or to monitor and evaluate community and staff responsiveness to such integration.

In this outbreak 2122 cholera cases and 44 deaths were reported, nearly all before the OCV campaign and over half occurred in the first 2 weeks of the outbreak. Of note, the outbreak started in February 2018 at about the same time the application for preventive use of OCV was being was submitted. The original application proposed a series of preventive campaigns over the next year, and Hoima, as well as neighbouring districts in western Uganda, were targeted for vaccination in the first round of these preventive campaigns. However, when the outbreak was identified, plans were quickly shifted so that an emergency campaign could be implemented to control the outbreak. Even though this emergency response was planned as quickly as possible, in fact, the outbreak was essentially over before the vaccine campaign could start, so it had no impact on the outbreak itself, but likely reduced the risk for future outbreaks.

Though the outbreak started with the influx of the refugees from DRC into Uganda, it quickly spread to the refugee host communities in Hoima. Therefore, to prevent rapid spread, improvement of cholera prevention measures for both the refugees and the host communities is paramount during resettlement.

## CONCLUSION

This study suggests that the OCV campaign in Hoima district to prevent cholera was successful and achieved a high level of coverage in this population at high risk. However, there was need to devote more effort on community sensitisation on the benefits of vaccination, as well as improving some logistic support during the campaign.

While a rapid response to this outbreak was appropriate, in fact, even with a rapidly organised campaign, the outbreak was over before the vaccine could be given; thus, the vaccine had no impact on this outbreak. Nevertheless, this area had already been identified as a hotspot, and it would have been targeted if the planned preventive campaign had proceeded as originally planned. Planners must realise that an area identified as a hotspot might experience an outbreak while preparations are underway for a preventive campaign and take this into account when allocating vaccine for preventive versus emergency campaigns. Since these are areas where cholera risk is high, outbreaks are likely to occur in these areas if there are delays in implementing preventive campaigns.

**Acknowledgements** The authors thank the persons and institutions who supported Hoima district OCV monitoring and evaluation study. Special thanks are extended to the research assistants who dedicated their time to meticulously collect the data and the persons who collaborated in the preparation of the different key documents used in this study. We thank the staff, the communities in the study area and the local leadership of Hoima districts for the cooperation during this study. We are grateful to Ambrose Buyinza Wabwire for technical support with drawing of the maps. We thank the Ministry of Health leadership for guidance; the Office of the Prime Minister (OPM) for the permission to access data on the refugee population. We are grateful to the following institutions and bodies; Ministry of Health, Makerere University School of Public Health, World Health Organisation, UNICEF and UNHCR for technical guidance.

**Contributors** Conceived and designed the study: GB, MR, WAB, AB, CGO and DAS. Discussed, critically revised and approved the study protocol: GB, MR, WAB, CGO and DAS. Performed the research: GB, MR, WAB, OA, FR and IA. Analysed the data: GB, MR, WAB, OA and DAS. Wrote the first draft: GB, MR. Wrote the final DAS. Elaborated, discussed and approved the final version: GB, MR, AB, AB, OA, FR, IA, CGO and DAS.

**Funding** Funding for the study was provided by the Bill and Melinda Gates Foundation (OPP1148763) which provided financial support through the John Hopkins University, Delivery of Oral Cholera Vaccine Effectively (DOVE) project.

**Disclaimer** The funding agency had no role in collecting, analysing or interpreting the results.

**Map disclaimer** The depiction of boundaries on this map does not imply the expression of any opinion whatsoever on the part of BMJ (or any member of its group) concerning the legal status of any country, territory, jurisdiction or area or of its authorities. This map is provided without any warranty of any kind, either express or implied.

**Competing interests** None declared.

**Patient and public involvement** Patients and/or the public were not involved in the design, or conduct, or reporting, or dissemination plans of this research.

**Patient consent for publication** Not required.

**Ethics approval** This study was conducted as part of the routine MoH operational research for improvement of health services; however, ethical issues were considered and addressed. The proposal was approved by the Makerere University School of Public Health Institutional Review Board (MaKSPH IRB) (no 610 in 2018) and Uganda National Council of Science and Technology.

**Provenance and peer review** Not commissioned; externally peer reviewed.

**Data availability statement** Data are available on reasonable request. Data can be made available by the Ministry of Health, Uganda through GB (cddmoh@yahoo.com).

**ORCID iD**
David A Sack http://orcid.org/0000-0002-9338-5119

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
