## [Reviewer comments · BMJ Open]

ARTICLE DETAILS

TITLE (PROVISIONAL)	Use of surveys to evaluate an integrated oral cholera vaccine campaign in response to a cholera outbreak in Hoima District, Uganda
AUTHORS	Bwire, Godfrey; Roskosky, Mellisa; Ballard, Anne; Brooks, W; Alfred, Okello; Rafael, Florentina; Ampeire, Immaculate; Orach, Christopher; Sack, David

VERSION 1 – REVIEW

REVIEWER	Justin Im International Vaccine institute Republic of Korea
REVIEW RETURNED	16-Jun-2020

GENERAL COMMENTS	Reviewer comments by section Abstract: First sentence awkward, consider revising Clarify how OCV was introduced following the outbreak – which population? What type of campaign? Use of “documentation” (Line 35) is vague, specify what information precisely was collected and is useful for ‘important lessons’ (Line 36) Rather than surveys ‘consisting’ (Line 37) of two sub-studies, authors could better describe two studies that were performed which utilized surveys for data collection, especially since it seems that the sampling techniques are different. Unclear if this is a qualitative or quantitative analysis Introduction: Typo ‘found’ (Line 111) □ ‘round’ Line 116 – incorrect use of semi-colon Line 118 – incorrect use of apostrophe Methods: Line 173: define ‘standard formula’ Is there any supporting evidence to support an average household size of 5? Provide. Is there any information on the total population of each cluster? What values of ‘p’ and ‘q’ were used to calculate the sample size?
--

	A clear definition of how vaccination status was ascertained is required. Line 195: was a person considered to be vaccinated only if they reported vaccination verbally AND this was indicated on their vaccine card? AEFI definition required. During what period after vaccination? Classified as serious? Results: Line 253: ‘Sixty seven percent...’ sentence unclear. Sub-study II: describe characteristics of clusters selected – mean, median size, rationale to support the notion that clusters are representative of the larger population (or not) Of vaccinations that were not confirmed by vaccination card, was this because the vaccination card was not available or because there was no record on the vaccine card – this is an important distinction and should be discussed. Table 3: definition of classification of mild, moderate, severe required. Status: ongoing, might be biased depending on the time after vaccination that the individual was visited. Was this addressed? Status: Improved, not to baseline – does baseline assume ‘healthy’? What age group was most impacted by AEFIs? Limited data presented on Community knowledge. Line 316: expand (RO). Was the model adjusted for any confounders? If so, state. Discussion: Authors claim that coverage may increase in more settle population without discussing the implications of the vaccination strategy or providing insight on whether other parts of Uganda indeed have more stable populations. Even in a highly mobile population, if the campaign is conducted within a short period of time, i.e. 1 month, and coverage is assessed directly afterwards, this may not impact the ability of the campaign to achieve high coverage. The authors seem to suggest that one of the high purposes of targeting high coverage is to ‘achieve herd protection’. This misses the point. Herd protection is a byproduct of vaccination influenced by the efficacy of the vaccine against the disease as well as the coverage of vaccination. High coverage, per se, is important since it is an indicator of the ability of the health system to reach the target vaccine population. Authors should also define ‘significant protection’ Line 328. Line 342: ‘previous OCV campaigns’ refer to those done in other countries? This is a weak comparison as it is a qualitative observation of vaccinations that occurred in different regions in different populations. Line 349: define ‘high coverage rate’. What point is the author trying to make here? Where other campaigns conducted in a similar fashion? Is cholera endemic in all other countries? Is there a reason to believe that the vaccine would not be accepted?
--	--

REVIEWER	Brecht Ingelbeen Institute of Tropical Medicine, Antwerp, Belgium
REVIEW RETURNED	28-Aug-2020

GENERAL COMMENTS	The manuscript describes a cholera outbreak and uptake survey following an OCV campaign in a district of Uganda bordering the DRC. Considering the 2017 WHO position to prioritize OCV use in cholera hotspots and during outbreaks, it is important to document uptake of, and learn from early OCV vaccination campaigns, as is done by the authors of this manuscript. The manuscript is clearly written. Study objectives, methods are clear, but some elements are missing in the methods to be able to assess potential bias and understand the validity of findings: especially a need to clarify who exactly was interviewed (household heads, household members of any age?), whether this was different for the different surveys undertaken, and some potential limitations to the sampling strategy (a village of 20 inhabitants was as likely to be selected as one of 1000 inhabitants?). Also, it is not fully clear how the conclusion is concluded from the results (which results support the 'need to devote more effort on community sensitisation on the benefits of vaccination...'?). Specific comments  - line 142: confusing subtitles, here and elsewhere in the methods. Difficult to search for specific information in the methods section. I would suggest to stick to more conventional structuring of methods: study population, sampling strategy, data collection, data analysis, ethics. - lines 170-71: who exactly were the study participants: all household members in selected households, only household heads? - lines 173-74: "standard formula for cluster sampling" -> it would be clearer for the reader to be explicit how you selected clusters and how you adjusted sample size for design effect (eg. what assumptions did you rely on?). Also it is unclear to me how clustering is accounted for in the formula in lines 174-178 - lines 180-181: see above comment. from this methods, it seems like someone in a smaller village is more likely to be selected than someone living in a large village. - line 186: 'number of households as indicated in Table 1' -> it seems to me from table one that that table specifies the number of individuals in each age category, not households. Are Table 1 the participants eventually interviewed? - line 192: household heads interviewed -> so not individual household members? did the household heads only respond for themselves, or for just one randomly selected household member? - lines 198-205: I would suggest to phrase this in a more simple way. eg. On top of routine passive reporting of ..., we... - line 219: why conditional log regression? - line 219: 'of uncommon events' -> unclear to me what you mean by this, and where these analyses are presented in the results - line 253: 66% of cases instead of 67% of cases. specify that % concerns cases. -lines 262-264: you mention KP1 and KP2, and then rounds 1 and 2. I assume round 1 refers to KP1? Good to be consistent in wording, to avoid misinterpretation. - line 275: unclear to me what you mean by this. - table 2: is quite messy, I would suggest reviewing the structure of this table, and specify whether clustering is accounted for here - line 292: n=? - line 300 - Table 3: what do you mean with 'treatment'?
---

	 - line 309 - Table 4: what is the use of having that many categories for the onset by each symptom? - lines 313-314: percentages and counts in the text could be helpful. - lines 355-361: if you've only been interviewing household heads, that could be a limitation. both potential selection bias (if these answer for themselves) as well as response bias (if they answer for other household members). - lines 363-363: you did clearly state that you were surveying knowledge and not attitudes, so to me this is not a limitation to your study - lines 389-390: how can this be concluded from the findings of your study?
--	--

VERSION 1 – AUTHOR RESPONSE

Comments of Review 1

There are two embedded sub-studies described in this article. Although both research activities describe a component of the vaccine campaign, they are unrelated to each other, and therefore distract from a unifying research objective. The described sub-studies assess different populations of interest and use different methods for collecting information for analysis.

We felt that it was important to include monitoring and evaluation of both the vaccine staff and the communities receiving the vaccine. Campaigns utilize a large number of staff to carry out the vaccine distribution, and this vaccine is quite different from other vaccines or interventions in which were involved. Also, logistically, this vaccine campaign is quite unique, so it seemed important to understand the constraints identified by the staff. Typically, campaigns report coverage rates without understanding the capacity and constraints of the campaign staff, but we felt that the two sub-studies were very much related.

Abstract ...because of the multiple methodologies and results included, the main take home messages are not clear.

We have revised the abstract significantly and hope the messages are more clearly stated.

The methods section, while lengthy, lacks clear, succinct, description of what was done.... More detail about the sample selection process is required as well as the inclusion of adjustment for potential confounding variables in statistical models

Although the methods section is somewhat lengthy, we felt these methods are important to describe in detail.

Regarding the sample selection, the methods section provides details for identifying the subjects in a representative manner. This methodology is quite similar to the standard 30 cluster surveys recommended by WHO.

Sections on ethical considerations and patient and public involvement can be reduced to contain the essential components.

We feel the description of the ethical considerations are needed to demonstrate that the study was carried out properly. The statement concerning public involvement was a requirement from the editor. The geographical scope and duration of the outbreak itself could be more clearly described as well as reporting of the cases and deaths...the authors may consider removing this section or inserting into the background.

We appreciate this suggestion and have moved the information of outbreak to the introduction. We do feel it is important information to include the epidemiological information about the epidemic curve since it gives context to better understand the outbreak and the need for the vaccine campaign.

Authors can consider substituting the order of substudy 1 and 2.

We feel this is an excellent suggestion and have changed the order of substudy 1 and 2.

Data from sub-study 1 (staff survey) would be better presented in a table.

I believe this information is best described in the text rather than a table as suggested. Although the staff were questioned about some specific items, the staff survey was more open ended and subjective in order to allow the staff to identify issues and constraints they faced and these responses would seem better suited to text.

Definitions of reported variables, such as AEFIs and categorization into mild moderate and severe should be explicitly stated or referenced.

The methods section defines AEFIs using fairly standard definitions as recommended by the FDA. We include a reference to these definitions. "They were considered mild if the symptoms did not interfere with normal activities; moderate if they interfered somewhat; and severe if the symptoms prevented the individual from continuing normal activities."

Were household respondents asked questions regarding the residents of the total household?

Yes, the informant for each household provided information about each member of the household.

This is clarified in the methods section.

When confirmation of vaccination status by vaccine card was unavailable, was this primarily because vaccine cards were not available or because vaccination was not indicated on the card?

The lack of a card to validate the vaccination was a combination of lack of a sufficient number of cards so they could not be provided to every subject and in some cases because the card was not available at the time of the visit. Documenting vaccination with cards is a common problem following a vaccine campaign, so is not unique to this campaign.

A closer look into more direct and clear alternatives to presenting data should be taken for all main results.

We have revised the results section substantially in hopes of addressing this comment.

Authors must provide more details on the methods of sample selection of clusters and state all assumptions made.... adjustment for confounders should be included in the methods

The methods for the selection of the clusters is provided in the methods section. This is a standard method for carrying out a two-stage cluster sample to obtain a representative sample of the population under study. We did not identify confounders that would be relevant when carrying out the cluster sampling.

The discussion section lacks form and clarity. Authors should begin by summarizing the main findings of the study and then citing limitations and strengths of the data, design and analyses performed. The impact of the study (public health, biological, political, etc) should be stated clearly. The limitations section of this study is not comprehensive and should be expanded significantly.

We appreciate this comment and have revised the discussion section considerably.

Reviewer 2

especially a need to clarify who exactly was interviewed (household heads, household members of any age?),

We clarified this in the methods. One informant provided information regarding all the members of the household. Generally, the informant was the head of the household, but sometimes was another responsible adult in the house.

Also, it is not fully clear how the conclusion is concluded from the results (which results support the 'need to devote more effort on community sensitization on the benefits of vaccination...?').

This was one of the items identified by the survey of the staff.

line 142: confusing subtitles, etc

By moving the epidemic curve to the introduction and switching the subgroups, we hope these subtitles are more clearly described.

lines 170-71: who exactly were the study participants

In our revision, we clarify that the study participants in the community survey include all members of the household.

lines 173-74: "standard formula for cluster sampling" - it would be clearer for the reader to be

explicit how you selected clusters and how you adjusted sample size for design effect (eg. what assumptions did you rely on?). Also it is unclear to me how clustering is accounted for in the formula in lines 174-178.

This was clarified in the revision.

Lines 180 -181: see above comment. From this methods, it seems like someone in a smaller village is more likely to be selected than someone living in a large village.

The number of villages selected in each sub-county was adjusted to the relative size of the sub-county. The individuals are assumed to represent the area, not the village.

Line 186: number of households as indicated in Table 1- it seems to me from table one that table specifies the number of individuals in each age category, not households. Are Table 1 the participants eventually interviewed?

Table 1 shows all the individuals in the household, but the information regarding them is obtained from the informant. This is clarified in the methods section.

Lines 198 – 205: I would suggest to phrase this in a more simple way. E.g. On top of routine passive reporting of...we...

The intent here was to suggest that the routine monitoring of AEFI during the campaign would detect only the illnesses sufficiently severe to being seen at a health facility. Our survey however was designed to detect symptoms that did not result in a visit to a health facility but did occur and need to be recognized. We did not attempt to compare our results with the results of the routine system.

line 219: why conditional log regression?

We had included plans to carry out this kind of analysis at the request of the IRB, but in fact, we did not find a need for conditional regression analysis and have removed this from the manuscript. This is now described as logistic regression.

line 219: 'of uncommon events' -> unclear to me what you mean by this, and where these analyses are presented in the results

This was suggested by the IRB in the protocol, but in fact there were no uncommon events and we have removed this.

line 253: 66% of cases instead of 67% of cases. specify that % concerns cases.

Thank you for this correction. This is now moved to the introduction section.

lines 262_264: you mention KP1 and KP2, and then rounds 1 and 2. I assume round 1 refers to KP1?

Good to be consistent in wording, to avoid misinterpretation.

We clarified this to mean that the KP1 and KP2 were the surveys among the staff that were conducted following vaccine rounds 1 and 2.

line 275: unclear to me what you mean by this

This sentence attempted to describe problems that the field staff incurred in carrying out their work.

In line 275, the workers expressed difficulty in reaching some households in more remote areas and they also expressed the need to have additional days to complete vaccinating all the households in the designated area. That is, they felt too rushed given the short amount of time available.

- table 2: is quite messy, I would suggesting reviewing the structure of this table, and specify whether clustering is accounted for here

Thank you for this suggestion. We have attempted to simplify this table.

line 292: n=?

here is our sentence now: "Among those who did not receive a dose of vaccine, over half of these missed doses (254 of the 357 missed doses during the two rounds) were because the person was not at home at the time of vaccination or was out of town. In a few cases, the vaccine team missed the household, accounting for 53 missed doses. Refusing to take vaccine was not reported."

line 300 - Table 3: what do you mean with 'treatment'?

We added this sentence to the methods section. “Among those reporting symptoms, information was recorded as to whether the person took any medicine or received any treatment to lessen the symptoms reported.”

line 309 - Table 4: what is the use of having that many categories for the onset by each symptom?

We reduced the number of categories in Table 4.

- lines 313-314: percentages and counts in the text could be helpful.

We reviewed this calculation and felt that a more conservative estimate should be provided. The new sentence is as follows. “There was a statistically significant association between education level and knowledge about OCV with those having at least a primary school education being almost twice as likely to know the number of required doses as compared to those with no education (OR 1.90, 95% CI 1.06, 3.44 [P = 0.03].”

Lines 355-361: if you've only been interviewing household heads, that could be a limitation, both potential section bias (if these answer for themselves) as well as a response bias (if they answer for other household members).

We clarified that a single informant provided information for all members of the household and we included this among the limitations of our study. We do not feel this is a major limitation however since the vaccine teams directly gave the vaccine directly to everyone in the household together and the informant was able to witness the others ingesting the vaccine.

Lines 363 : you did clearly state that you were surveying knowledge and not attitudes, so to me, this is not a limitation of your study.

We do feel this is an important point suggesting that future work is needed to understand attitudes that are important to controlling cholera in an integrated manner.

lines 389-390: how can this be concluded from the findings of your study.

This refers to this sentence: “However, there was a need to devote more effort on community sensitization on the benefits of vaccination, as well as improving some logistic support during the campaign.” This statement is based on the results of the staff survey.

FORMATTING AMENDMENTS (if any)

- You have indicated ‘Yes’ to this question. With this, please indicate the number/ID of the approval(s).

This is included in the manuscript.

- The author “Ampeire, Immaculate” in your main document is registered as “Ampaire, Immaculate” in ScholarOne. Please ensure that the author has same registered name.

Ampeire is the correct name. I will correct this.

Attached review

Reviewer comments by section (assume this is reviewer #3)

Abstract:

First sentence awkward, consider revising. Clarify how OCV was introduced following the outbreak – which population? What type of campaign? Use of “documentation” (Line 35) is vague, specify what information precisely was collected and is useful for ‘important lessons’ (Line 36) Rather than surveys ‘consisting’ (Line 37) of two sub-studies, authors could better describe two studies that were performed which utilized surveys for data collection, especially since it seems that the sampling techniques are different. Unclear if this is a qualitative or quantitative analysis.

We completely revised the abstract according to the format of the journal.

Introduction:

Typo 'found' (Line 111) . 'round'

Corrected, thank you.

Line 116 – incorrect use of semi-colon .

Line 118 – incorrect use of apostrophe

These were corrected

Methods:

Line 173: define 'standard formula'

This is now explained in the manuscript.

Is there any supporting evidence to support an average household size of 5? Provide.

The number was provided through information from a recent DHS survey. In fact, the observed family size in this study was somewhat larger, 6.02 individuals per household (not including infants less than one).

Is there any information on the total population of each cluster?.

There were 31 clusters with 209 households, with a total population of 1274 individuals.

What values of 'p' and 'q' were used to calculate the sample size?

We used 50% for these values. This is now stated in the text.

A clear definition of how vaccination status was ascertained is required. Line 195:

We added this sentence: Vaccination status was ascertained in two ways: either the informant verbally indicated that the individual had received the dose of vaccine or the vaccination was recorded on the vaccine card.

AEFI definition required. During what period after vaccination? Classified as serious?

There were no serious AEs, but the symptoms described as mild, moderate, or severe are based on the definitions as described in the text. We added a sentence to indicate that there were no serious AEs.

Results:

Line 253: 'Sixty seven percent...' sentence unclear.

Sentence revised.

Sub-study II: describe characteristics of clusters selected – mean, median size,

There were 7 households in each cluster but the number of clusters per sub-county was proportionate to the population of the sub-county. The mean number of individuals in a household was 6.1.

rationale to support the notion that clusters are representative of the larger population (or not)

This is a standard method for obtaining a representative sample using cluster surveys and is recommended by WHO for EPI coverage surveys.

Of vaccinations that were not confirmed by vaccination card, was this because the vaccination card was not available or because there was no record on the vaccine card – this is an important distinction and should be discussed.

As explained above, the lack of card verification resulted from a combination of not having enough numbers of cards to provide to all vaccine recipients and the lack of availability of the card at the time of a visit. The lack of cards to verify 100% of the vaccinations is a common finding in such studies.

Table 3: definition of classification of mild, moderate, severe required. Status: ongoing, might be biased depending on the time after vaccination that the individual was visited. Was this addressed?

Status: Improved, not to baseline – does baseline assume 'healthy'?

Definitions are provided in the methods section and are those recommended by the FDA. A reference is provided.

Baseline does not assume healthy, but it does refer to the status of the individual prior to receiving the vaccine.

What age group was most impacted by AEFIs?

Only abdominal pain was sufficiently frequent to possibly break down by age group. Do we have this information? Of 15 reported cases of abdominal pain: >5 years=1, 5-14 =4, >14=10). It would seem that young children would not express this symptom, so an age breakdown may not be relevant. I note that 7 of these were in females of reproductive age – possibly menstrual cramps??

Limited data presented on Community knowledge. Line 316: expand (RO). Was the model adjusted for any confounders? If so, state.

As stated above, upon reanalysis of this data, we decided to use a more conservative estimate of this association. The sentence now reads, “There was a statistically significant association between education level and knowledge about OCV with those having at least a primary school education being almost twice as likely to know the number of required doses as compared to those with no education (OR 1.90, 95% CI 1.06, 3.44 [P = 0.03].” We did not adjust for confounders.

o

Discussion:

Authors claim that coverage may increase in more settled population without discussing the implications of the vaccination strategy or providing insight on whether other parts of Uganda indeed have more stable populations. Even in a highly mobile population, if the campaign is conducted within a short period of time, i.e. 1 month, and coverage is assessed directly afterwards, this may not impact the ability of the campaign to achieve high coverage.

It would seem that persons living in a more settled population could be more efficiently identified and followed up if they were not present during a first visit.

The authors seem to suggest that one of the high purposes of targeting high coverage is to ‘achieve herd protection’. This misses the point. Herd protection is a byproduct of vaccination influenced by the efficacy of the vaccine against the disease as well as the coverage of vaccination. High coverage, per se, is important since it is an indicator of the ability of the health system to reach the target vaccine population.

We agree that herd protection is affected by many factors, but there is considerable evidence regarding OCV that herd protection can result from the level of coverage obtained in this campaign. Authors should also define ‘significant protection’ Line 328.

We added two references to support the claim for protection for persons not receiving vaccine by way of herd protection.

Line 342: ‘previous OCV campaigns’ refer to those done in other countries? This is a weak comparison as it is a qualitative observation of vaccinations that occurred in different regions in different populations.

We agree that there are limitations to this observation; however, it is reassuring that nearly all campaigns in different countries have reported high coverage rates. The methods used in the different studies are not identical, but at least none have reported low coverage.

Line 349: define ‘high coverage rate’. What point is the author trying to make here? Where other campaigns conducted in a similar fashion? Is cholera endemic in all other countries? Is there a reason to believe that the vaccine would not be accepted?

We attempted to explain that there could be problems when a country introduces oral cholera vaccine for the first time, especially an emergency campaign in the middle of an outbreak. This is a new vaccine given to all age groups. It is given orally unlike most vaccines and for a disease for which vaccines have never been given. Not only is this vaccine new to the communities, but it is also new to the field staff. We attempted to explain these concerns in our revision.

VERSION 2 – REVIEW

REVIEWER	Brecht Ingelbeen Institute of Tropical Medicine Antwerp, Belgium
REVIEW RETURNED	09-Oct-2020

GENERAL COMMENTS	Thanks for considering the comments from the first review. I think the paper advanced well, and is complete and very informative now. There's one element that I would have liked to see in the data analysis section: how the multistage sampling (clustering) has been accounted for in estimating the percentages. Few more minor suggestions: -why do you not provide 95% confidence intervals with the coverage estimates in the text? it is a sample of the population which was surveyed, while you infer the results of that sample to the population, so there is some uncertainty, isn't it?- I would suggest to clarify you assume selected households within the same subcounty are likely to be similar regardless of which village they live in (as clarified in the rebuttal), since village size was considered in the sampling strategy- line 433: "targeted to be at risk" you mean targeted because they are at risk?
--

VERSION 2 – AUTHOR RESPONSE

Reviewer: 2

There's one element that I would have liked to see in the data analysis section: how the multistage sampling (clustering) has been accounted for in estimating the percentages.

The households interviewed for the study provide a representative sample of the households in the catchment area. The use of a two-stage cluster sample is a logistically convenient method to obtain a representative sample of the population as described in reference #17.

-why do you not provide 95% confidence intervals with the coverage estimates in the text? it is a sample of the population which was surveyed, while you infer the results of that sample to the population, so there is some uncertainty, isn't it?

We have added the 95% confidence intervals in the text and in the revised abstract.

- I would suggest to clarify you assume selected households within the same subcounty are likely to be similar regardless of which village they live in (as clarified in the rebuttal), since village size was considered in the sampling strategy

The cluster sampling methodology assumes that the overall data obtained from the selected households will be similar to the entire catchment population. We did attempt to determine if there are differences between sub-counties or other geographic sub-groups. We did however enlarge our sample size to evaluate potential difference by sex.

- line 433: "targeted to be at risk" you mean targeted because they are at risk?

Thank you for noting this. I have changed this sentence to be ... "achieved a high level of coverage in this population at high risk."

VERSION 3 – REVIEW

REVIEWER	Brecht Ingelbeen Institute of Tropical Medicine Antwerp, Belgium
REVIEW RETURNED	10-Nov-2020

GENERAL COMMENTS	The article improved, especially its structure and clarity of several substudies. However, in the data analysis subsection in the methods, it is not specified how prevalences and their confidence intervals were calculated, and how the two-stage sampling and clustering, are taken into account. It probably will not have affected results much, but it would be important to know if this is adjusted for.
---

VERSION 3 – AUTHOR RESPONSE

The reviewer is correct that it would have been better if we had adjusted the confidence limits for the cluster. Unfortunately, we did not record the cluster number so were not able to make this correction. I described this limitation in the discussion section. This limitation does not affect the estimates of the mean values we report, but it would affect the variance slightly. I believe this limitation may be partly compensated by our oversampling, but i did not include this statement in the discussion.